# Computed Tomography-Derived Occipital–Coccygeal Length and Ilium Wing Distance Correlates with Skin to Epidural and Intrathecal Depths in Dogs

**DOI:** 10.3390/vetsci7040196

**Published:** 2020-12-03

**Authors:** Tsim Christopher Sun, Mariano Makara, Fernando Martinez-Taboada

**Affiliations:** 1Department of Anaesthesia, The University Veterinary Teaching Hospital, School of Veterinary Science, The University of Sydney, Sydney, NSW 2050, Australia; fernando.martinez@sydney.edu.au; 2Diagnostic Imaging Unit, The University Veterinary Teaching Hospital, School of Veterinary Science, The University of Sydney, Sydney, NSW 2050, Australia; mariano.makara@sydney.edu.au

**Keywords:** epidural, intrathecal, lumbosacral, sacrococcygeal, anaesthesia, computed tomography, ligamentum flavum

## Abstract

The current clinical techniques for neuraxial needle placement in dogs are predominantly blind without prior knowledge of the depth required to reach the desired space. This study investigated the correlation and defined the relationship between easily obtainable external landmark variables in the dog; occipital–coccygeal length (OCL) and ilium wings distance (IWD), with the skin to epidural and intrathecal space distances using computed tomography (CT). The CT images of 86 dogs of different breeds were examined in this retrospective observational study. Images of dogs in sternal recumbency were optimized to the sagittal view. The distances between the skin and lumbosacral epidural space (LSE) and skin to sacrococcygeal space (SCE) were measured to the ligamentum flavum surrogate (LFS) line. The distance between the skin and the intrathecal space (ITS) was measured from the skin to the vertebral canal at the interlumbar (L_5_–L_6_) space. Measurements of the IWD and OCL were performed on dorsal and scout views, respectively. Linear regression equations and Pearson’s correlation coefficients were calculated between variables. Data were reported as mean (standard deviation). Significance was set as alpha < 0.05. After exclusion of four dogs, 82 CT scans were included. The depths were LSE 45 (15) mm, SCE 23 (10) mm, and ITS 50 (15) mm. There was a moderate correlation between OCL with LSE (=14.2 + OCL * 0.05 (*r* = 0.59, *p* < 0.0001)), and a strong correlation with ITS (=11.4 + OCL * 0.07 (*r* = 0.76, *p* < 0.0001)), while a very weak correlation was found with SCE (=14.0 + OCL * 0.02 (*r* = 0.27, *p* < 0.0584)). Similarly, with IWD, there was a moderate correlation with LSE (=10.8 + IWD * 0.56 (*r* = 0.61, *p* < 0.0001)), and strong correlation with ITS (=9.2 + IWD * 0.67 (*r* = 0.75, *p* < 0.0001)), while a weak correlation was found with SCE (=11.2 + IWD * 0.2 (*r* = 0.32, *p* < 0.0033)). Mathematical formulae derived from the multiple regression showed that the body condition score (BCS) improved the relationship between IWD and OCL and the LSE, SCE and ITS, while the addition of body weight was associated with multicollinearity. Further studies are required to determine the accuracy of the algorithms to demonstrate their ability for prediction in a clinical setting.

## 1. Introduction

Procedural failure rates for neuraxial anaesthesia in dogs have been reported to be up to 32% for epidural [1], and 29.4% for spinal anaesthesia [2]. Failure to access the desired space results in a variety of complications, ranging from inadequate analgesia to potentially fatal total spinal anaesthesia should the intended volume of epidural drug be injected intrathecally [3]. Most clinicians perform these techniques without any radiographic assistance, so they are blindly selecting the needle trajectory and the final target. In these circumstances, they must rely on subjective tactile and audible sensations such as the ‘pop’ of the ligamentum flavum to signal the entry into the epidural space, or the visualization of either the hanging drop or the free flow of cerebral spinal fluid to confirm the correct identification of the epidural or the intrathecal space, respectively [2].

Additional knowledge of the required depth and the factors affecting needle placement might be beneficial in increasing operator accuracy and improving the success rate and safety of epidural and spinal anaesthesia [4]. In humans, anthropometric variables affecting the distance from the skin to the lumbar epidural space have been investigated [4,5,6,7,8]. Body mass index and body weight showed higher levels of correlation compared with age, gender and height with the lumbar epidural space in a population of Nigerian adults [9]. In another study, positive correlations were found for body weight, body mass index and body surface area with the same space in Greek men, with differences further seen for obstetric and non-obstetric populations of Greek women [4]. Both studies produced mathematical models to predict the skin to lumbar epidural space distance in their respective populations, and while there were questions regarding the clinical utility of the derived formulae, potential usefulness may lie in a multimodal approach with other techniques, as well as an initial guide for needle placement for less experienced operators.

In dogs, there are limited studies investigating the skin to epidural depth. Only one study investigated the correlation of weight and body condition score with skin to epidural space distance, finding a moderate and weak correlation, respectively [10]. Unfortunately, in that study, the clinical efficacy of the hanging drop technique was not tested, and it was unknown whether successful epidural anaesthesia was achieved. Alternatively, the distance between the skin and other clinically relevant neuraxial structures, such as the sacrococcygeal epidural space and intrathecal space, has not been evaluated, despite being of common interest for several veterinary specialities, such as anaesthesia, neurology, internal medicine, and radiology. It therefore seems prudent and logical to find clinically relevant variables and investigate the potential ability for correlations and predictions of epidural or intrathecal space depth prior to the actual attempt.

The palpation of external landmarks is routinely performed for neuraxial techniques to guide the clinician towards the site of interest, such as palpating the craniodorsal aspects of the iliac wings to identify the lumbosacral space [3]. Other external measurements may not currently be common practice but are gaining popularity, such as calculating the volume of epidural injectate based on the occipital–coccygeal length [11,12]. These relationships between external anatomical landmarks and internal structures have not been fully explored in dogs, but it is plausible that they closely correlate with the skin to epidural and intrathecal space distances.

The aim of this study was to assess the correlation between external anatomical landmarks (occipital–coccygeal length, ilium wing distance), and skin to epidural and intrathecal space depths using CT images, with dog body condition score and body weight. We hypothesized that the measurements obtained would demonstrate a close correlation between all variables.

## 2. Materials and Methods

### 2.1. Case Selection

This was a retrospective review of dog computed tomography (CT) images and hospital records. Abdominal, lumbar or whole-body CT scans of dogs obtained under anaesthesia as part of the individual clinical management between September 2018 and June 2020 at the University Veterinary Teaching Hospital Sydney, Australia, were included. The scans were performed using a 16 slice multidetector CT scanner (Phillips 16 Slice, Brilliance CT V2.3; Phillips Medical Systems, Eindhoven, The Netherlands) with a 2 mm transverse slice thickness and 512 × 512-pixel matrix dimensions. The scanning protocol at the institution has been described previously [13]. All owners consented to the procedure and general anaesthesia, and, as per the institutional policies, consented to future data use. Dog details, including body condition score (BCS out of 9, with 9 being obese [14]), body weight, and sex (male or female), were retrieved from the patient file or the anaesthetic record on the day of the CT. Leg positioning was recorded from the CT studies and defined as cranial (pelvic limbs extended forward), neutral (pelvic limbs to the side) or caudal (pelvic limbs extended backwards). Dogs were excluded if they were not in sternal recumbency, had incomplete studies, had fractures or obvious distortion of the spine or pelvis, movement artefact, or anatomical deformity such as hemivertebrae or a lack of clearly defined coccygeal vertebrae.

### 2.2. Distance Measurements

All images were analysed by a single observer (TCS) and viewed on three dimensional reconstructions using a bone Hounsfield unit window (Apple Thunderbolt Display, Apple, Cupertino, CA, USA; Mac Mini, Apple; Osirix version 5.7 64-bit, Pixmeo SARL, Bernex, Switzerland). All measurements were made within the software through the bone algorithm from the sagittal plane unless otherwise specified and converted to the nearest millimetre.

#### 2.2.1. Ligamentum Flavum Surrogate

A straight line was drawn from the cranial margin of the dorsal lamina of the cranial lumbar space vertebrae, to the cranial margin of the caudal vertebrae (i.e., from L_5_ to L_6_). This was defined as the ligamentum flavum surrogate (LFS).

#### 2.2.2. Intrathecal Space Depth (ITS)

The ITS was measured as the distance from the skin to the floor of the vertebral canal using a line drawn perpendicular to the LFS at the L_5_ and L_6_ intervertebral space, and represented the theoretical distance the tip of the needle should travel before reaching the ventral intrathecal or subarachnoid space (ITS: Figure 1a).

#### 2.2.3. Lumbosacral Epidural Space Depth (LSE)

The LFS was then drawn at the lumbosacral space, and a perpendicular line was drawn. This line was defined as the LSE90. A secondary line was then drawn to the skin with an angle of 60 degrees to the LFS intersection [15], and was defined as the LSE60 (LSE: Figure 1b). Both the LSE90 and LSE60 represented two possible angular approaches to the LFS. The skin to LFS distances were measured for both angular approaches.

#### 2.2.4. Sacrococcygeal Epidural Space Depth (SCE)

The LFS process was repeated at the first to second inter-coccygeal space to represent the sacrococcygeal epidural space due to differences in the curvature of the sacral vertebrae, and the secondary line was measured for the distance at 60° to the skin and defined as the SCE (Figure 1c).

#### 2.2.5. Ilium Wing Distance (IWD)

The most dorsal aspects of the ilium wings were identified on the dorsal plane window where both bony structures were first clearly visible. A straight line was drawn, and the distance was measured between the cranial margins of the dorsal crests (IWD: Figure 1d).

#### 2.2.6. Occipital–Coccygeal Length (OCL)

The OCL was taken from the scout view which was visually optimised to the dorsal line. The open polygon function was used to measure the distance between the base of the occipital bone to the first coccygeal vertebrae following the outline of the dog’s body (OCL: Figure 1e).

### 2.3. Statistical Analysis

A sample size of 82 dogs was calculated to be needed based on four predictors with a power of 0.8 and an alpha of 0.05 [16]. Measurements were recorded in a computerized spread sheet (Microsoft Excel 2011; Microsoft Corporation, Redmond, WA, USA), and transferred to the statistical software R, version 3.6.1 for Windows 10 (The R Foundation for Statistical Computing, http:www.Rproject.org, Vienna, Austria). Statistical analysis was performed in three stages and in all sections statistical significance was declared at alpha less than 0.05.

The first analysis involved consolidation of data as appropriate and descriptive statistical analysis. The LSE60 was used as a standardised comparison between leg positions (cranial, neutral, and caudal), sex (male, female), and angulation with LSE90. Data were examined for normality using the Shapiro–Wilks test. When normal data distribution was found, a Student’s *t*-test was used. If data were non-normally distributed, non-parametric tests were used. Where there was no statistical significance between groups, further analysis was reported as a consolidated group for leg positions and sex, and for angulation the LSE60 was used for further analyses and redefined as the LSE. If statistical difference existed, then groups were reported individually. Descriptive statistical analysis for continuous data—body weight, LSE, SCE, ITS, IWD, and OCL—were expressed as mean (standard deviation). For categorical data, binomial classification of BCS was performed (less than or equal to BCS 5/9, and greater than BCS 5/9). The reason for the classification was to reduce the subjectivity of a categorical scoring system, and to be able to examine the addition of BCS in the multivariate model.

The second analysis investigated the relationship between depth variables (LSE, SCE, ITS), external landmarks (IWD, OCL), and BCS and body weight. Depth variables were the outcome and estimates were performed by producing linear regression equations and Pearson’s correlation coefficients with external variables, BCS and body weight. Correlation coefficients were further calculated between external variables, BCS and body weight. Results were expressed as positive or negative correlation values of Pearson’s *r*, where values between 0 and 0.29, 0.3 and 0.49, 0.5 and 0.69, and 0.7 and 1.0 were considered to signify very weak, weak, moderate and strong correlations, respectively. Relationships were examined graphically using crude values, and the linearity of residuals and fitted values were assessed in R. Normality was assessed using Q-Q plots and the D’Agostino–Pearson Omnibus test. Results were further expressed as β coefficients to examine the weight of each variable, *_adj_R*^2^ for the proportion of fit, F statistic and the associated statistical significance for the predictability variable for the outcome. In the third analysis, the best models were selected for the relationship of depth and external variables with the addition of BCS or body weight using multiple regression by determining the model that had the lowest variance inflation factor.

## 3. Results

A total of 86 canine CT scans were examined. Four dogs with incomplete scout views were excluded from analysis, providing a total of 82 dogs for the final analysis. There were 44 dog breeds identified (Appendix A: Table A1). For BCS, there were 47 dogs with BCS ≤ 5/9 (BCS 1/9 = 0; 2/9 = 2; 3/9 = 5; 4/9 = 18; 5/9 = 22), and 35 dogs that had BCS > 5/9 (BCS 6/9 = 18; 7/9 = 11; 8/9 = 6; 9/9 = 0) (*p* < 0.0001). The overall body weight was 19.9 (15.1) kg and the LSE, SCE and ITS were 44 (15), 23 (10), and 50 (15) mm, respectively. The measured OCL was 589 (173) mm and IWD was 61 (16) mm. There were no differences when examining leg positions which were either in neutral (*n* = 43) or caudal (*n* = 39) *(p* = 0.0924), and no dogs had their legs in a cranial position. There were also no differences between males (*n* = 42) and females (*n* = 40) (*p* = 0.7760), and angulations (*p* = 0.2148).

### 3.1. Correlation between Depth Variables Using External Variables

When correlating depth with external variables, the OCL had a moderate correlation with LSE, and a strong correlation with ITS, while the correlation between OCL and SCE was very weak. This generated the equations:(1)LSE = 14.2 + OCL∗0.05 (r = 0.59, 95% CI 0.43–0.71; p < 0.0001)
(2)ITS = 11.4 + OCL∗0.07 (r = 0.76, 95% CI 0.65–0.84; p < 0.0001)
(3)SCE = 14.0 + OCL∗0.02 (r = 0.27, 95% CI 0.05–0.46; p = 0.0584)

The IWD also had a moderate correlation with LSE, and a strong correlation with ITS, while a weak correlation was found between IWD and SCE. This generated the equations:(4)LSE = 10.8 + IWD∗0.56 (r = 0.61, 95% CI 0.45–0.73; p < 0.0001)
(5)ITS = 9.2 + IWD∗0.67 (r = 0.75, 95% CI 0.63–0.83; p < 0.0001)
(6)SCE = 11.2 + IWD∗0.20 (r = 0.32, 95% CI 0.11–0.50; p < 0.0033)

### 3.2. Correlation between Depth Variables, Bodyweight and BCS

Body weight had a strong correlation with LSE (*r* = 0.75, 95% CI 0.64–0.83, *p* < 0.0001), a strong correlation with ITS (*r* = 0.85, 95% CI 0.78–0.90, *p* < 0.0001), and a weak correlation with SCE (*r* = 0.47, 95% CI 0.28–0.62, *p* < 0.0001). For BCS, there was a moderate correlation with LSE (*r* = 0.53, 95% CI 0.35–0.67, *p* < 0.0001) and a weak correlation with ITS (*r* = 0.36, 95% CI 0.16–0.54, *p* = 0.0009). There was a moderate correlation between BCS and SCE (*r* = 0.64, 95% CI 0.49–0.75, *p* < 0.0001).

### 3.3. Correlation between External Variables, Body Weight and BCS

The OCL strongly correlated with body weight (*r* = 0.88, 95% CI 0.82–0.92, *p* < 0.0001), but no correlation existed with BCS (*r* = 0.002, 95% CI −0.22–0.22, *p* = 0.9835). Similarly, IWD strongly correlated with body weight (*r* = 0.87, 95% CI 0.81–0.92, *p* < 0.0001), but no correlation existed with BCS (*r* = −0.005, 95% CI −0.22–0.21, *p* = 0.9081).

### 3.4. Multiple Regression Analysis

The addition of BCS improved the *_adj_R*^2^ and the relationship between the external landmarks and depth variables. Table 1 shows the effects of external, BCS, and bodyweight relationships on depth. Furthermore, a variance inflation factor of 1 suggests BCS and external variables did not possess multicollinearity. Alternatively, when body weight is added to the model, inversely related or statistically insignificant β coefficients suggest that variables may not be independent of one another and do not explain the improved *_adj_R*^2^ in the multiple regression model. Multicollinearity was detected between body weight and external variables. The six mathematical formulae derived for this were:(7)LSE=7.3+0.05(OCL)+16.45(BCS)
(8)SCE=8.4+0.02(OCL)+13.3(BCS)
(9)ITS=6.8+0.06(OCL)+10.88(BCS)
(10)LSE=3.5+0.56(IWD)+16.6(BCS)
(11)SCE=5.3+0.20(IWD)+13.34(BCS)
(12)ITS=4.38+0.67(IWD)+11.05(BCS)
Variables: LSE, SCE, ITS, OCL and IWD in mm; BCS, ≤5/9 = 0, >5/9 = 1

Using Equation (7) as an example, the depth of the LSE can be predicted based on OCL and BCS. A significant regression equation was found (F = 65.86 *p* < 0.05) with an *_adj_R*^2^
*=* 0.62 (Table 1). The predicted LSE depth is equal to 7.3 + 0.05(OCL) + 16.56(BCS), where BCS ≤ 5/9 = 0, >5/9 = 1, and OCL is measured in millimetres. The LSE depth increased by 0.05mm for each mm of OCL, and dogs with a BCS > 5/9 had an LSE 16.45 mm greater than dogs with BCS ≤ 5/9. Both OCL and BCS were significant predictors of LSE. Details of univariate relationships can be found in the Appendix A (Table A2) and multiple regression analysis in Table 1.

## 4. Discussion

The mean measurement of the LSE, SCE and ITS in our study was 45, 23, and 50 mm, respectively, from the skin. Dogs that had a longer OCL and wider IWD had moderate and strong correlations with LSE and ITS, respectively, while the correlation with SCE was weak. Dogs that were heavier or had a BCS of >5/9 had larger LSE, SCE and ITS compared to lighter dogs, or dogs with BCS ≤ 5/9. There was no difference between male or female dogs, the angulation of the LSE line at 60 or 90 degrees, or between neutral and caudal leg positions on CT.

The measurements of LSE, SCE and ITS in this study were a result of CT-derived distances as determined by an artificial line to represent the ligamentum flavum. Comparisons to other studies are therefore difficult given differences in the study aim and outcome. Only two other studies have reported on the distance from the skin to the lumbosacral epidural space in dogs, with a mean depth of 26.8 mm reported by Iseri et al. [10], and 59.5 mm by da Silva et al. In this latter study, the authors only investigated the skin to lumbosacral epidural depth in obese Labradors [17]. The distance from the skin to the lumbosacral epidural space reported in our study is noticeably greater than that reported by Iseri et al. Unfortunately, the methodology of both studies was very different, making any comparison difficult. For instance, Iseri et al. identified the epidural space by using the hanging drop, a technique that has been suggested to be insufficiently accurate (in terms of sensitivity and specificity) for research [18]. Additionally, the authors did not provide information regarding the success rate of their epidural techniques (e.g., reporting the number of animals requiring rescue analgesia in the perioperative period). Finally, the skin to epidural space distance was measured by grasping the Tuohy needle with a pair of forceps as close to the skin as possible. Then, the reported distance was measured from the needle tip to the forceps. It is possible that the authors might have unintentionally depressed the skin while grasping the needle, obtaining underestimated results.

Our study found positive moderate and strong correlations between OCL and IWD and the LSE and ITS, respectively, while both external variables had a weak correlation with SCE. Neither the OCL nor IWD have previously been investigated with regard to their relationship to the skin to epidural or intrathecal space depth. The OCL has primarily been used for the determination of injectate volume into the epidural space [11], but to the authors’ knowledge, the iliac wings have been used exclusively as landmark palpation sites of the lumbosacral space in dogs and there is no information regarding any alternative role the IWD may have in practice. Our results may serve as an anchor point for future clinical studies in predicting the skin to epidural and intrathecal space depth using external variables, and whether the knowledge of that information can have an effect on the outcomes of the technique (rate of success and complications). Moreover, the results reported here may be of use for other aims than neuraxial anaesthesia; for instance, the strong correlations with ITS might make them of great interest while performing myelograms or cerebrospinal fluid sampling [19,20].

Our data also confirmed that these positive correlations between OCL and IWD and the LSE, ITS and SCE can be improved with the inclusion of BCS. In the univariate analysis, there was a moderate correlation between the LSE and BCS. Our results also show that this correlation is slightly stronger at the level of the SCE, possibly due to greater accumulation of fat deposits at the base of the dog’s tail as BCS increases [14]. Iseri et al. reported a weaker correlation between the skin to epidural space depth and BCS (*r* = 0.26 (*p* = 0.004)) than the one reported in our study [10]. The comparisons between studies are difficult given differences in BCS classification used (Iseri et al. used a 1 to 5 classification, whilst a 1 to 9 classification was used in this study), the subjectivity of the assessment of BCS and study population. The effect of body condition has also been investigated in human studies using body mass index, with correlations also being of a positive direction, possibly due to increased subcutaneous fat between the skin to the desired space [5,8,9].

Our study also showed that weight had positive strong correlations with LSE and ITS, and weak with SCE. A similar relationship was found in the study by Iseri et al. for weight and the skin to the lumbosacral epidural space (*r* = 0.78, *p* < 0.01) [10]. However, in the multiple regression, body weight was an overall poorer fit for LSE and SCE, and similarly for the ITS when compared with BCS. In addition, multicollinearity was detected with greater variance inflation factors (Table 1). This suggested that collinearity existed between variables, in which independent relationships are actually dependent on each other. A collinear relationship reduces the accuracy of the model and removal of the variable should be considered for further multiple regression analyses [21]. In the presence of external variables such as OCL and IWD, the use of body weight may therefore be inferior to BCS in the prediction of the outcome. Similarly, studies in human anaesthesia have shown that many factors may influence body weight as a variable in predicting skin to lumbar epidural space depth, for example ethnicity [8] or sex [4].

Differences were not observed between sex or leg positioning, although only dogs with legs in neutral and caudal positioning were found in the database. Most radiographers prefer the pelvic limbs of the animals to be extended caudally or resting neutrally so as to not increase beam hardening and noise artefact when acquiring the images. A previous study investigated the effect of limb positioning on the canine lumbosacral anatomy using CT, demonstrated an increase in the lumbosacral and L_6_–L_7_ distances, when the limbs are extended cranially versus caudally [22]. It is possible that the limbs’ position might also influence the skin to epidural and intrathecal space depths. The effect of pelvic limb position might also be an explanation for the difference in the LSE reported in our study and the one reported by Iseri et al. [10]. Further prospective studies should consider the pelvic limbs in a cranially extended direction (given that the majority of epidural procedures in dogs are performed in such a fashion).

The angles used in this study from the skin to the LFS were divided into perpendicular from the skin (LSE90) and the 60 degrees angle (LSE60). Unpublished pilot results from our research group show that the angles for successful lumbosacral epidural needle placement ranged between 60 and 90 degrees, so it was decided to use the extremes of the range. The methodology used for ITS and SCE followed the previously described techniques for needle placement into the sacrococcygeal [12] and intrathecal space [23]. When the measurements obtained for LSE90 and LSE60 were compared, no significant difference was found. It is paramount to point out that distances to any of the studied spaces using different trajectories (e.g., when utilizing the paramedian approach to the intrathecal space), or with angles outside the ones proposed may differ from the ones obtained in this study.

All the measurements in this study were taken from the CT images. This may be of limited clinical value given that a scan would be required to determine the LFS and the relevant depth variables prior to performing the relevant techniques. However, this might not be that uncommon as the use of CT is becoming more popular and accessible in veterinary practice, and the images might have previously been acquired for related or unrelated medical reasons [24].

The skin to epidural distance measured in this study was obtained using the surrogate LFS. The LFS was created as a theoretical representation of the ligamentum flavum. Anatomically, the ligamentum flavum is not a rigid structure but stretches segmentally between intervertebral spaces [3]. Unfortunately, this ligament is not usually visible on CT in dogs [25], so a representative straight line had to be used to allow a standardised objective comparison between sites of interest. In human adults, the posterior aspect of the spinal cord [26], and the anterior longitudinal ligament at corresponding inter-spinous spaces [27], have been previously used as CT surrogates for entry into the thoracic epidural space. While theoretical in nature, the surrogate was an objective and replicable way to investigate the depth variables and their correlations with other variables of interest.

This study had other limitations derived from its retrospective nature. All the dogs were in sternal recumbency, as this is the most common positioning for this type of radiographic study. Scans with the animals in dorsal are not common in our institution, but occasionally this recumbency might be used to avoid breathing movement artefacts (e.g., CT scans of the spine). They were all excluded, as it was thought that that would greatly impact on the skin-to-epidural distance. Moreover, the dog breeds were reported, but not included in our analyses. The main interest of this study was to correlate external morphometric variables with depth variables. It is reasonable to think that these morphometric variables capture at least some of the breed conformations variations [28,29,30,31]. Any attempt to stratify the results by breed would have resulted in extremely underpowered results. Perhaps, a future study with a large sample size might be able to shed further light on this point.

## 5. Conclusions

In conclusion, external variables such as the OCL and IWD correlate positively with LSE, SCE and ITS, and when BCS is also considered, this improves the model. Further investigations are required before this information can be used clinically to assess the skin to epidural space and skin to spinal space depth, prior to attempting the needle placement. Firstly, the LFS must be compared to epidurography to corroborate its validity. Secondly, the morphometric variables values obtained by CT must be compared to the external measuring of such variables, so they can be obtained without the need for advance imaging.

## Figures and Tables

**Figure 1 vetsci-07-00196-f001:**
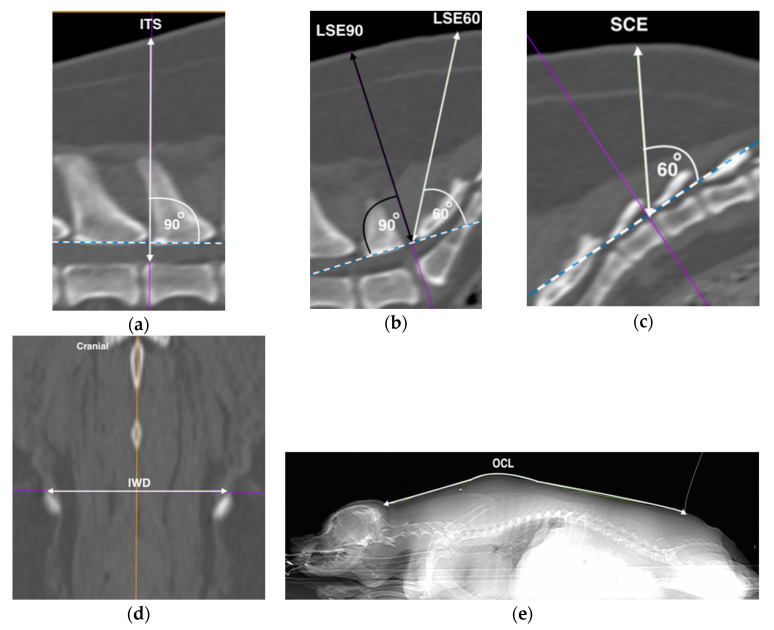
Computed tomographic images representative of measurements performed. Hashed line represents the ligamentum flavum surrogate (LFS), with the acute angle depicted by the unidirectional arrow perpendicular to the LFS. Bi-directional arrows represent measured distances of skin to: (**a**) intrathecal (ITS), (**b**) lumbosacral epidural at 90 degrees (LSE90) in black, and 60 degrees (LSE60) in white, and (**c**) sacrococcygeal epidural (SCE) spaces. (**d**) The distance between the most dorsal aspects of the iliac wings (IWD), and (**e**) the occipital–coccygeal length (OCL) on the dorsal and scout windows, respectively. Left is cranial unless specified. All images were from the same dog.

**Table 1 vetsci-07-00196-t001:** Predictors of lumbosacral epidural (LSE), sacrococcygeal epidural (SCE), and intrathecal (ITS) space depth with external and patient variables, using computed tomography of 82 dogs by multiple regression analysis.

Variable		LSE					SCE					ITS				
External	Dog	β_external_	β_dog_	*_adj_R* ^2^	F_stat_	VIF	β_external_	β_dog_	*_adj_R* ^2^	F_stat_	VIF	β_external_	β_dog_	*_adj_R* ^2^	F_stat_	VIF
OCL	BCS	0.0524 *	16.4511 *	0.6156	65.86 *	1.0000	0.0156 *	13.2870 *	0.4681	36.64 *	1.0000	0.0658 *	10.8791 *	0.7008	95.87 *	1.0000
	KG	−0.0286 *	1.0613 *	0.5807	57.1 *	4.3715	−0.038 *	0.0709 *	0.2997	18.33 *	4.3715	0.0042	0.8082 *	0.7226	106.5 *	4.3715
IWD	BCS	0.5594 *	1.6594 *	0.643	73.93 *	1.002	0.1979 *	13.3353 *	0.5048	42.29 *	1.003	0.6662 *	11.0524 *	0.6811	87.49 *	1.002
	KG	−0.1926	0.9605 *	0.5675	54.14 *	4.1717	−0.2270	0.5419 *	0.2352	13.46 *	4.1716	0.0034	0.8473 *	0.7221	106.2 *	4.17

OCL: Occipital–coccygeal length; IWD: ilium wing distance; BCS: body condition score; KG: body weight in kilograms; β: standardised coefficients of the variable in subscript within the multiple regression model; *adjR*^2^: adjusted coefficient of proportion of fit; F_stat_: test for overall significance in regression; VIF: variance inflation factor test for multicollinearity; * statistically different for the respective variable (*p* < 0.05), otherwise no statistical difference detected.

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
