# Peer review of "Computed Tomography-Derived Occipital–Coccygeal Length and Ilium Wing Distance Correlates with Skin to Epidural and Intrathecal Depths in Dogs"

_vetsci, 2020, doi:10.3390/vetsci7040196_

Round 1

Reviewer 1 Report

Dear authors:

Thanks for improving your manuscript.

Good luck

Reviewer 2 Report

After the revision of the article, I still struggle to see the benefit of this study. I agree that similar studies have been performed in human medicine, however, many different breeds of dogs exist with a large variety in anatomy (chondrodystrophic vs non-chondrodystrophic dogs, dogs with a high incidence of vertebral malformations such as brachycephalic dogs). If one would like to examine if external morphometric variables would potentially help to estimate the depth of the epidural space, this should be examined in one breed and not a variety of breeds.

Even though the authors added a part in the discussion regarding the position of the hind limbs, this is a major flaw of this study for which cannot be corrected.

Reviewer 3 Report

The authors didn't answer my 2 questions.

Reviewer 4 Report

The response to my comments is unsatisfactory.

The results of this work show a correlation between morphometric variables, first of all the BCS, and the distance between skin and epidural space. This result has no clinical importance, because it is logical that in an obese dog the adipose panniculus is greater than in a thin dog. This work does not bring any additional information nor is it of any help to the clinician. The skin-intervertebral foramen distance can be identified with ultrasound (Etienne AL et al, 2010). And other authors have already demonstrated the correlation between morphological parameters and volume of anesthetic to be inoculated (Valverde A, 2019)

Reviewer 5 Report

I don't have any further comments

This manuscript is a resubmission of an earlier submission. The following is a list of the peer review reports and author responses from that submission.

Round 1

Reviewer 1 Report

The manuscript present novel data about epidural and spinal space distances from the skin and their correlations to external anatomical measures based on retrospectively collected canine CT images.

The manuscript is clearly written and well presented, limitations have been correctly addressed, I don't have any suggestions for improvement. 

Reviewer 2 Report

Work well structured and well written; excellent intuition and technique that allows measurements with a tomographic method, which is certainly safer than others performed in veterinary medicine

Two general considerations:

  1. it is a job that must be useful to the clinician, in a particular circumstance for which the distance and the distance to be completed assume great importance, in the lumbar space, especially for young doctors with relative experience. Therefore, the average of the measurements is useless and misleading: out of 84 breeds there is a variation from Chihuaha to Rhodesian ridgeback, also because the results speak of a relationship with the BCS. It should also be verified whether the correlation on the population is always present within the individual size groups. The measures of the spaces, the real purpose of such a work and which would be really interesting for the clinic, are not there.
  2. the authors speak of previous works saying that they have the limitation of not having provided clinical data, not having verified the possible result of lumbar anesthesia. This clinical evidence should also have been verified.

Reviewer 3 Report

Dear Authors,

First of all congratulations for your manuscript, however I have some questions:

  1. How many times the single observer measured each part that was analysed? 
  2. Why you didn't have a second observer to analyse and make the same  mesurements ?? 

Reviewer 4 Report

The article describes multiple measurements that were retrospectively performed on CT scans to determine distances between skin and epidural and intrathecal depths in dogs. Correlations between external landmarks and BCS and body weight were investigated.

The current study does not have any added value. Although intrathecal depths are evaluated at the level of L5-L6, this is clinically irrelevant for at least all dogs being > 15 kg as the spinal cord terminates in these dogs at the level of L6. Body fat is not equally distributed over the body and typically more fatty tissue is found more caudally on the back in dogs (lumbosacral region). Consequently, correlations found in this study are not necessarily the same if these would be looked at in different areas in the back. In order to palpate the ilium wings the easiest to place an epidural needle, legs are often placed in a cranial position; none of the dogs in this study underwent a CT scan in this position. Finally, mainly in fatty dogs, landmarks are often very difficult to palpate making it difficult to know were the needle ideally needs to be introduced in the skin, rendering the distance the needle needs to covered vary variable.

Reviewer 5 Report

Comments to the authors of the manuscript : Computed Tomography Derived Occipital-Coccygeal Length and Ilium Wing Distance Correlates with Skin to Epidural and Intrathecal Depths in Dogs

 The manuscript investigates several measurements and characteristics of dogs in correlation to the depth between skin and epidural or intrathecal space. Multiple statistical tests are performed to support the conclusions.

 Line 175 and following: What does the second number in brackets mean?

Line 257: That could be reasonable. I think, however, that the present study included all measurements obtained from CT scan- except the BCS. In this way, the clinical significance of the project drops drastically, because in these patients, the LSE, SCE and ITS can be directly measured from the CT study as needed. As mentioned in the last lines, to render this data valid, a correlation between CT measures and dog should have been investigated. I recommend to prospectively investigate that in a smaller group of patients or in dogs cadavers.

Line 304: Another limitation of the study is that the reported measurements refer only to the dorsoventral direction. In case of injections for clinical purposes, the exact needle position in the other 2 dimensions (in the latero-lateral and cranio-caudal) have to be established as well.

Line 325: It has to be specified in the entire manuscript, that the external morphometric variables are obtained by CT images.

Reviewer 6 Report

Reviewer comments on Manuscript number: VetSci-931353

The present manuscript shows the correlation between external anatomical landmarks, and skin to epidural and intrathecal space depths using computed tomography images, with dog body condition score and body weight in dogs

The manuscript is well written, and the information showed is interesting and relevant, as its main aim is to find useful data for a prediction in a clinical setting.

 Broad comments

The study is well designed, and the data showed is valuable.

Weakness of the study: Lack of breed stratification.

Specific Comments

Abstract

L 29. Please define BCS.

Introduction

L 39-40. The paragraph is not clear. Please rephrase.

Materials and Methods

L 132-133. The arrow colors are inverted (LSE90 black and LSE60 White)

L 148. Please specify, if appropriate, the non-parametric test used.

Discussion

L 241. Please change to passive voice “….studies was different”

Thank you.